# An optimised protocol for the detection of lipofuscin, a versatile and quantifiable marker of cellular senescence

**Camilla S. A. Davan-Wetton[1], Trinidad Montero-Melendez[1,2]***

**1** The William Harvey Research Institute, Faculty of Medicine and Dentistry, Queen Mary University of London, London, United Kingdom, **2** Centre for Inflammation and Therapeutic Innovation, Queen Mary University of London, London, United Kingdom

* t.monteromelendez@qmul.ac.uk

## Abstract

Lipofuscin is a yellow-brown pigment typically found in the lysosomes that contains a mixture of molecules including lipids, metals and misfolded proteins. The use of Sudan black B to detect lipofuscin accumulation, a well described marker of cellular senescence and ageing, was first described in 2013 by Georgakopoulou, *et al*. Here, we provide an optimisation of the original protocol. Firstly, we adjusted the staining methodology for increased ease of use on cultured cells. Secondly, we show that Sudan black B-stained lipofuscin emits strong fluorescence in the far-red channel making it suitable for fluorescence microscopy detection and quantification. Moreover, we also demonstrate that this optimised protocol can be utilised in conjunction with standard immunofluorescence staining techniques, making possible the simultaneous detection of lipofuscin and other cellular proteins of interest, like additional markers of senescence. This is a significant advantage over the most commonly used method for senescence detection, based on beta galactosidase enzymatic activity. We therefore believe that these findings and the provided optimised protocol will represent a useful tool for the scientific community in the field of cellular senescence.

## Introduction

Lipofuscin is a yellow-brown complex substance that accumulates over time mainly in the lysosomes of post-mitotic cells [1]. The structure cannot be fully defined as it is a heterogeneous mixture of oxidized proteins, lipids, carbohydrates and metals such as iron, copper, zinc and others. Due to its highly cross-linked structure, lipofuscin cannot be degraded by the proteasome nor eliminated by exocytosis. Healthy young cells, however, are able to dilute their lipofuscin content during every round of cell division [2]. For this reason, senescent cells, i.e. cells in permanent cell cycle arrest, build-up large amounts of lipofuscin. Equally, other post-mitotic cells like neurons also accumulate lipofuscin, which can be used as a biomarker of cellular ageing and even to estimate the absolute age of the organism [3]. "Ageing pigment" is thus a term typically used to refer to lipofuscin [4].

**Funding:** This research was funded by a Barts Charity Research Project Grant (G-002392) awarded to TMM. CSADW was funded by a Biotechnology and Biological Sciences Research Council (BBSRC) LIDo PhD studentship (BB/M009513/1). The funders had no role in study design, data collection and analysis, decision to publish, or preparation of the manuscript.

**Competing interests:** The authors have declared that no competing interests exist.

The formation of lipofuscin is believed to be mediated by reactive oxygen species, whereby oxidation of proteins results in their misfolding and aggregation [5]. Mitochondria are also believed to play an important role in lipofuscin formation [6], a theory that nicely links lipofuscin accumulation with lysosomal homeostasis and ageing. Non-functional mitochondria are eliminated by mitophagy, a process involving the lysosomal system [7]. Ageing is associated with the acquisition of damaged mitochondria, which increases oxidative stress, together with a reduced lysosomal capacity, causing the accumulation of this "cellular waste" [8]. Lipofuscin, in turn, affects cellular proteostasis mechanisms by directly inhibiting the proteasome pathway, resulting in a vicious cycle whereby impaired proteasome activity and lipofuscin accumulation reinforce each other. Eventually, lipofuscin accumulation can result in cytotoxicity leading to cellular death by apoptosis [9]. It is unsurprising, therefore, that lipofuscin elimination has been proposed as a potential anti-ageing strategy [10].

Extensive research on lipofuscin has been conducted in the context of the central nervous system and related pathologies. Lipofuscin deposits are associated with multiple conditions, for example dementia, Parkinson's disease, Alzheimer's disease, Huntington's disease, and a group of fatal diseases known as neuronal ceroid lipofuscinoses [11, 12]. Due to its natural autofluorescent nature, lipofuscin deposits have been detected in tissue sections, revealing not only the increased presence of these aggregates during ageing and the above pathologies, but also that certain regions of the brain seem to be resistant to lipofuscin accumulation [13]. The pathogenic effects of lipofuscin are also well documented in macular degeneration where lipofuscin photosensitivity becomes toxic for retinal pigment epithelium cells by enhancing oxidative stress [14]. Although lipofuscin is mainly contained in the lysosomes, it can also leak into the cytoplasm, or even be secreted. Chromhidrosis is a rare condition in which patients secrete coloured sweat, typically dark blue, derived from the accumulation of lipofuscin in apocrine glands [15].

More recently, Georgakopoulou *et al.* reported that lipofuscin is a marker of replicative and γ-irradiation induced senescence, and that these aggregates co-localize with senescence-associated β-galactosidase (SA-β-Gal) activity, using *in vitro* cultured primary fibroblasts and cancer cell lines as well as cryo-preserved tissue sections [16]. The development and improvement of methods to detect reliably and quantify lipofuscin may help to better understand the pathways leading to its formation, as well as what role it plays, for example in cellular senescence, which to date is largely unexplored. While the natural fluorescence of lipofuscin allows its detection in tissues, this can lack specificity and may be affected by high background autofluorescence in tissue sections [11]. Another disadvantage of relying on autofluorescence is the large spectrum of emission ranging from 400nm to 700nm, which varies depending on the tissue and prevents its co-localisation with other cellular components by immunofluorescence as it interferes with several channels. In addition, the autofluorescence levels are often too weak to be detected in cells cultured *in vitro*. Histochemical staining methods to detect lipofuscin have been developed, like staining with Sudan black B dye, which detects the lipid component of lipofuscin, and periodic acid Schiff staining, which detects the carbohydrate portion [16, 17]. For instance, Sudan black B has successfully been used to detect lipofuscin in a wide variety of cell and tissue types, including lung fibroblasts, osteosarcoma cells and prostate hyperplasia tissues [16], epithelial cells [18], ovarian interstitial gland cells [19] or cardiomyocytes [20].

Here we present an optimised, step-by-step protocol to detect the presence and allow the quantification of lipofuscin *via* Sudan black B (SBB) staining, using cultured human primary fibroblasts. This protocol provides an easier alternative to coverslip-grown cells [18] and can be used for brightfield and, as a novelty, for fluorescence microscopy, as we noted that staining lipofuscin granules with SBB makes them highly fluorescent in the far-red region (Cy5 channel) while simultaneously quenching lipofuscin natural autofluorescence in the green

fluorescent protein (GFP) and red fluorescent protein (RFP) channels. Importantly, we also demonstrate that the SBB staining protocol presented here can be used in conjunction with antibody-based immunofluorescence standard procedures, allowing the simultaneous detection of lipofuscin and other proteins of interest.

## Materials and methods

### Primary human dermal fibroblasts

Human dermal fibroblasts from healthy volunteers were kindly provided by Professor David J Abraham and Professor Christopher Denton (University College London, UCL) and were obtained from 4mm skin punch biopsy specimens from the forearm, isolated as described by Shi-wen *et al.* 2021 [21]. The study was approved by the NRES Committee, London-Hampstead, Health Research Authority, Research Ethics Committee London Centre, reference 6398, and all subjects gave written informed consent. Donors had an average age of 60 years (± 6 years) at time of donation, and both male and female donors were used. Fibroblasts were maintained in Dulbecco's modified eagle medium (DMEM; ThermoFisher, 31053028) supplemented with 10% heat inactivated foetal bovine serum (FBS; ThermoFisher, 10500064), 2 mM L-glutamine (Sigma, 59202C), 100 U/ml penicillin and 100 μg/ml streptomycin (Merck, P4333), known as 'complete medium', until ~80% confluence. Cells were always sub-cultured at 1:3 split using 0.05% trypsin-0.02% EDTA solution (ThermoFisher, 25300062) for 5 minutes, before neutralization with complete DMEM. Cells were used between passages 2–14. To obtain fibroblasts under replicative senescence, cells were maintained for >21 passages.

### Chemical compounds and treatments

The compounds used in this study to induce senescence were: BMS-470539 (Tocris, 4053) used at concentration range of 1–20μM, methotrexate (Merck, A6770) used at 1μM, bleomycin (Cayman Chemicals, CAY13877) used at 3.3μM and hydrogen peroxide (Fisher Scientific, H/1750/15) used at 50μM. Dimethyl sulfoxide (Merck, 76855) was used as the vehicle to dissolve bleomycin. All other compounds were dissolved in Dulbecco's phosphate buffered saline pH 7.1–7.5 (PBS, Merck, D8537). In all cases, cells were treated with the compounds for a total of six days and compounds added on days 1,3 and 5 after plating. After that, media was removed, and the cells washed with PBS and subjected to the different staining procedures.

### Sudan black B (SBB) staining

The protocol described in this peer-reviewed article is published on Protocols.io (https://dx.doi.org/10.17504/protocols.io.x54v9yw91g3e/v1) and is included for printing as S1 File with this article. Briefly, a saturated SBB solution was prepared by dissolving 1.2g of Sudan black B (Merck, 199664) in 80ml of 70% ethanol, with the solution stirred overnight and filtered prior to use. Cells were fixed in 4% paraformaldehyde in PBS (PFA, Santa Cruz, sc-281692) for 15 minutes, before rinsing in 70% ethanol for 2 minutes. Cells were then incubated in saturated SBB solution for 8 minutes on an orbital shaker set to 200 revolutions per minute (rpm). After the staining period, cells were washed for 5 minutes in distilled water. All washes were performed on an orbital shaker at 200 rpm. Lipofuscin accumulation was then visualised using a brightfield microscope (EVOS XL Core Imaging System, ThermoFisher) and a fluorescence microscope (EVOS Fl Imaging System equipped with Cy5 Light Cube (far red, EX628/40—EM685/40), ThermoFisher Scientific).

## Senescence-associated β-galactosidase (SA-β-Gal) staining

For the histochemical detection of SA-β-Gal activity in cultured cells, the Senescence Detection Kit (Abcam, ab65351) was used according to manufacturer's instructions. Cells were fixed with the provided fixative for 15 minutes. Staining solution containing 1mg/ml 5-bromo-4-chloro-3-indolyl-β-D-galactopyranoside (X-Gal) was then added and cells incubated in sealed plates at 37˚C overnight (~16 hours). After that period, the staining solution was removed, cells washed with PBS and the development of blue colour indicative of SA-β-Gal activity visualized using a brightfield microscope (EVOS XL Core Imaging System, ThermoFisher).

## Senescence-associated β-galactosidase (SA-β-Gal) and Sudan black B (SBB) co-staining

To co-stain for both markers, the SA-β-Gal procedure explained above was applied first as it depends on enzymatic activity and requires the use of fresh cells. Then, to co-stain with SBB, cells were washed once in PBS and subjected to the SBB staining protocol, performing steps 8–12, as in S1 File and Protocols.io (https://dx.doi.org/10.17504/protocols.io.x54v9yw91g3e/v1). As previously, after the staining period, cells were washed for 5 minutes in distilled water, and visualised using a brightfield microscope (EVOS XL Core Imaging System, ThermoFisher).

## Immunofluorescence staining

Cells were fixed in 4% PFA (Santa Cruz) for 20 minutes at room temperature before the fixative was removed and the cells washed thrice for 10 minutes each in PBS. Cells were then incubated in 5% normal goat serum (Abcam, ab7481) diluted in PBS for 1 hour followed by incubation in primary antibody (mouse anti-human alpha smooth muscle actin, clone 1A4, DAKO, M0581, at 1:200 dilution) overnight at 4˚C. Cells were then washed thrice for 10 minutes each in PBS before incubating in secondary antibody (Alexa Fluor 594 conjugated goat anti-mouse IgG (H+L), ThermoFisher, A-11032, at a dilution of 1:200) for 2 hours at room temperature. The cells were then washed twice in PBS for 10 minutes each, followed by a 10-minute incubation in a 1µg/ml DAPI solution in PBS (Merck, D9542). The cells were then washed for a final 10 minutes in PBS before visualisation using the EVOS FL Imaging System (ThermoFisher). After imaging, the cells were counterstained with SBB solution (commencing at Step 8 in the protocol provided in S1 File and Protocols.io (https://dx.doi.org/10.17504/protocols.io.x54v9yw91g3e/v1), i.e. incubation in SBB solution) and then visualised using then EVOS FL Imaging System (ThermoFisher).

## Fluorescence microscopy visualization and quantification

Cells were visualised using the EVOS FL Imaging System (ThermoFisher) at different channels depending on the experiments: Cy5 (far red, EX628/40—EM685/40) was used to detect SBB-stained lipofuscin; Texas Red (red, EX585/29—EM628/32) was used to detect Alexa Fluor 594 conjugated antibodies; DAPI (blue, EX357/44—EX447/60) was used to detect nuclei and the channels GFP (green, EX470/22—EM525/50), RFP (orange EX531/40—EM593/40) were used to detect autofluorescence levels. To quantify the fluorescence signal of SBB-stained cells, Cy5 images were imported into the open-source Fiji imaging processing package (ImageJ2, version 2.14.0/1.54f) and the Analyze/Measure/Integrated Density function used to quantify the fluorescence intensity of the whole image. The fluorescence intensity was then normalised by the

number of cells and expressed as fluorescence intensity per cell. Images were also analysed to express the data as percentage of positive cells.

## Data analysis

The value of n is defined as the number of technical replicates, as this work aims to determine the technical reliability of the procedure presented. Statistical parameters including the exact n value for each experiment, meaning of error bars, and statistical significance are reported in the figure legends. Data are considered statistically significant when p < 0.05. One-way ANOVA or Pearson's correlation tests were used as appropriate and indicated in each figure legend. In the figures, asterisks denote statistical significance (*p< 0.05; **p< 0.01; ***p< 0.001, ****p< 0.0001). Statistical analysis was performed in GraphPad Prism v10.

## Results

### An improved protocol for SBB staining of in vitro cultured cells

A method to detect lipofuscin as a marker of cellular senescence in cultured cells using Sudan black B dye has previously been reported [16]. However, the protocol required the use of coverslips, with tedious and meticulous steps needed to avoid dye precipitation over the cells, like performing the staining placing the samples upside down [18]. Here we provide an improved, fast protocol that can be easily performed in standard cell culture plastic plates. All the details for this procedure, materials and reagents are provided in S1 File and are also available through the protocol management platform Protocols.io (https://dx.doi.org/10.17504/protocols.io. x54v9yw91g3e/v1). Briefly, cultured cells are washed in PBS and then fixed in 4% PFA for 15 minutes. Next, the cells are rinsed in 70% ethanol before incubation in a saturated solution of Sudan black B for 8 minutes. As shown in S1 Fig, we found that performing this step using a plate shaker completely prevents the formation of precipitates that may confound the visualization of lipofuscin aggregates and avoids the need to use inverted coverslips. Another critical factor is the use of a freshly prepared Sudan black B solution, as heavy precipitation can occur even under shaking conditions when using an old solution (S2 Fig). The procedure concludes by washing the cells in distilled water. Lipofuscin can then be visualized using a standard brightfield microscope, and, as will be detailed later, using a fluorescence microscope equipped with a far-red laser (Cy5, EX628/40—EM685/40). Counter-staining with Nuclear Fast Red or with DAPI can also be performed to visualize cell nuclei under a brightfield or a fluorescence microscope, respectively (Fig 1).

### Lipofuscin aggregates are a pan-marker of cellular senescence

Previously, it was shown that lipofuscin accumulates in senescence cells derived from telomere shortening, γ-irradiation and overexpression of various cell cycle regulators like p21[WAF-1] and p53 [16]. Here, using our new improved protocol for lipofuscin staining, we show that senescence induced by several chemical compounds is also associated with lipofuscin accumulation. Human primary fibroblasts were treated every other day for 6 days with DNA-damaging agents like methotrexate (1μM) and bleomycin (3.3μM), with oxidizing agents like hydrogen peroxide ($H_2O_2$, 50μM) and with the melanocortin compound BMS-470539 (10μM), all known to induce distinct senescence phenotypes. In addition, cells grown for at least 21 passages, reaching a replicative senescence stage, were also used. Although with different cellular distribution, lipofuscin aggregates were easily identifiable after SBB staining in cells treated with all the pro-senescence compounds tested (Fig 2),

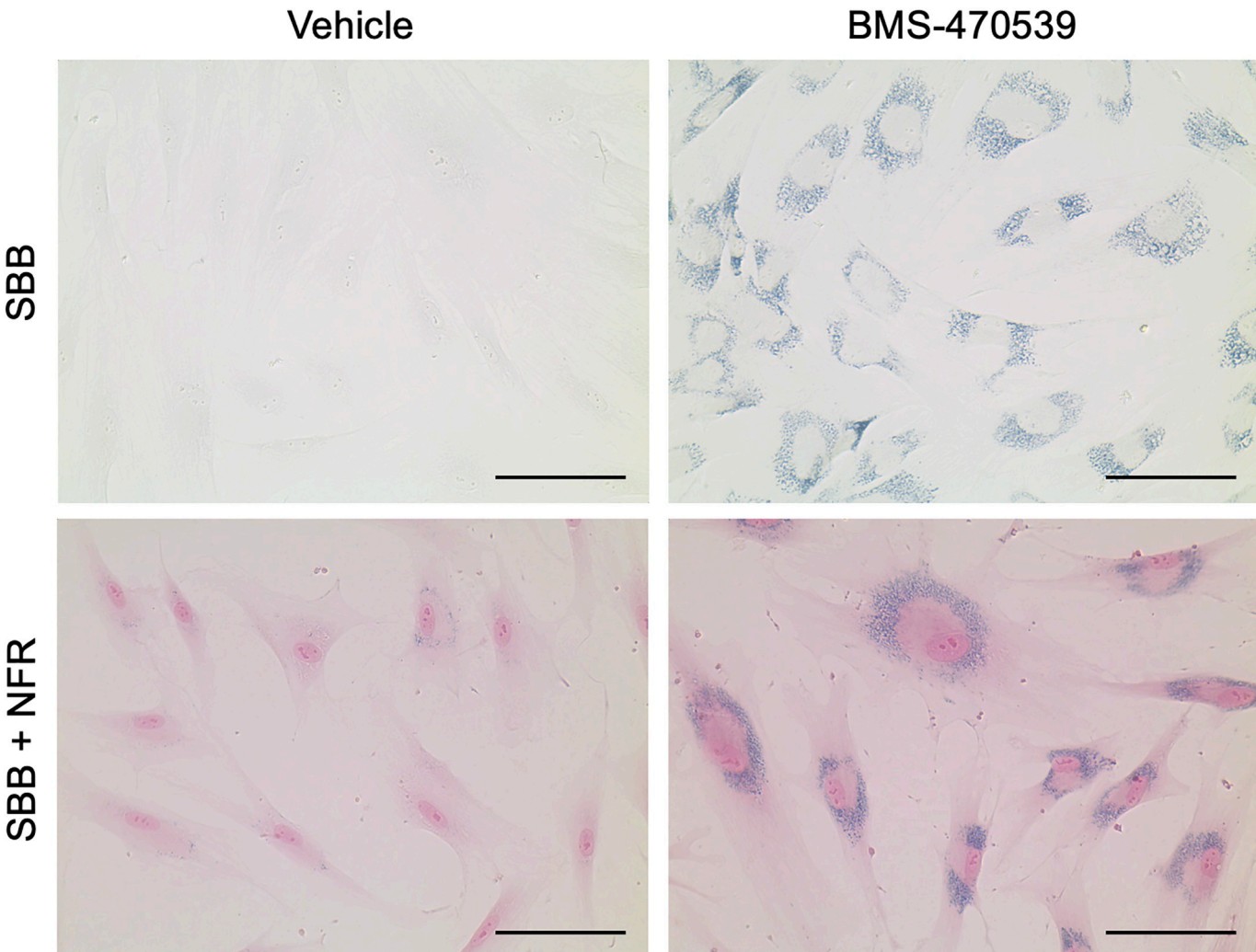

**Fig 1. Staining of lipofuscin using Sudan black B dye.** Healthy dermal fibroblasts were treated with 10μM BMS-470539 or vehicle (PBS) every other day for 6 days. Cells were then subjected to the staining protocol described in this work and visualised using an EVOS XL Core Imaging System at 40X magnification. The top panel corresponds to cells stained only with Sudan black B (SBB) and the bottom panel corresponds to cells stained with SBB and counter-stained with Nuclear Fast Red (NFR) to visualise the nuclei. Scale bars are equal to 100μm.

while absent in vehicle treated cells (PBS and DMSO). As expected, we also observed lipofuscin formation in cells undergoing replicative senescence. Similarly, SA-β-Gal activity, the most commonly used marker of senescence used in *in vitro* cultured cells, was also present in all types of senescence and clearly co-stains with SBB. However, detection of SA-β-Gal requires extensive optimization of staining times, as the visualization of SA-β-Gal enzymatic activity is context dependent and largely varies across senescence types, as exemplified in Fig 3. Importantly, we demonstrate here that applying the same protocol and timing (i.e. 8 minutes incubation time with SBB solution) allows the reliable and specific detection of lipofuscin in all cases. These data suggest that lipofuscin formation is a universal feature of cellular senescence, including DNA damage, stress- and drug-induced senescence, and can be easily used as a biomarker to detect this cellular process using the same protocol.

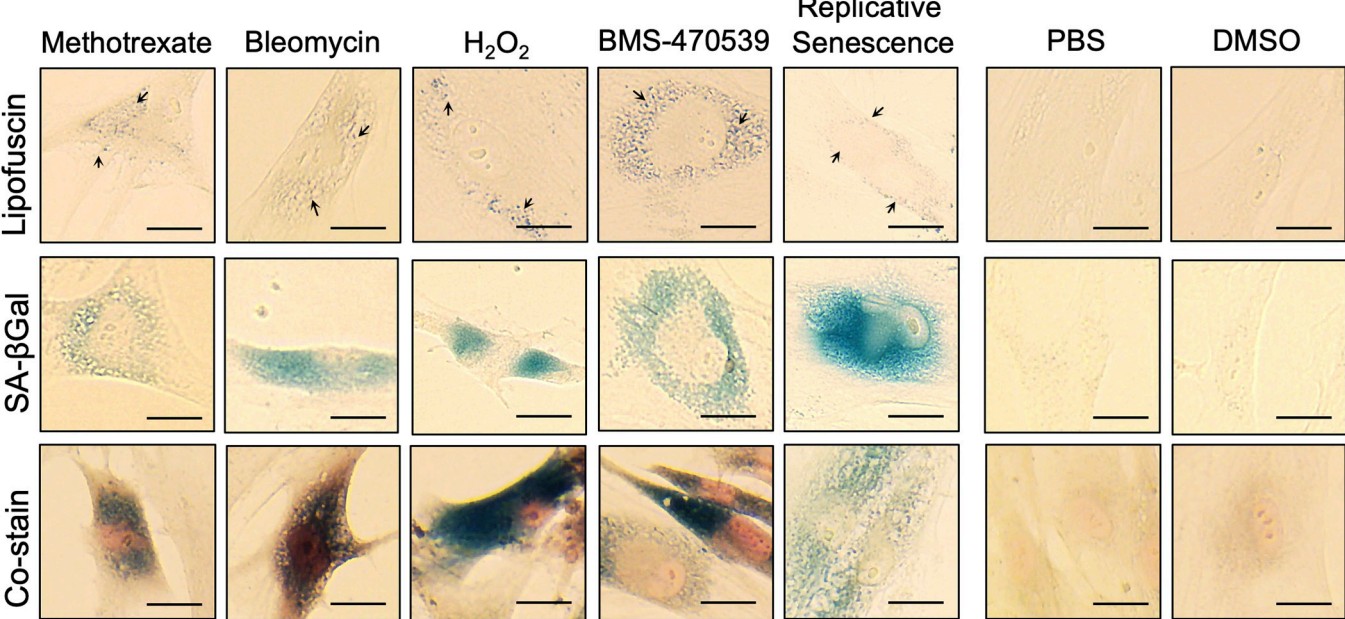

**Fig 2. Co-staining of lipofuscin with SA-β-galactosidase and Sudan black B.** Healthy dermal fibroblasts were treated with corresponding vehicle (PBS or DMSO), 10μM BMS-470539, 1μM methotrexate, 50μM hydrogen peroxide (H$_2$O$_2$) or 3.3μM bleomycin every other day for 6 days. To obtain cells under replicative senescence, these were sub-cultured for >21 passages. At the end of the treatment period, cells were stained for lipofuscin accumulation with SBB (top panel), SA-β-Gal activity (middle panel) and co-stained for SA-β-Gal activity followed by SBB staining (bottom panel). Images were captured using an EVOS XL Core Imaging System at 40X magnification. Scale bars are equal to 25μm. Arrows indicate lipofuscin aggregates.

## SBB-stained lipofuscin emits fluorescence in the far-red channel Cy5

In the previous experiments we identified that lipofuscin aggregates accumulate after treatment with various pro-senescent chemical agents. As the strongest response was obtained for the small molecule BMS-470539, the rest of the experiments presented here were conducted using primary fibroblasts treated every other day with 1, 5, 10 or 20μM BMS-470539 for 6 days, where drug was added on days 1,3 and 5 as previously. The natural autofluorescence of lipofuscin is well documented (12). As expected, lipofuscin was detected at day 6 using the fluorescent channels GFP (green, EX470/22—EM525/50) and RFP (orange, EX628/40—EM685/40) (Fig 4A). This, however, required high exposure times (~750 milliseconds) to be detectable. SBB staining (Fig 4B) effectively quenched the fluorescence emitted at those channels (GFP and RFP), as expected, because this chemical is typically used as a quencher of background signal in other fluorescence analytical techniques [22, 23]. However, we unexpectedly observed that when cells were stained with SBB, the lipofuscin aggregates were easily identifiable using the far-red Cy5 channel (EM628/40—EX685/40) (Fig 4B), providing a new opportunity for the detection of lipofuscin in cultured cells, with improved sensitivity compared to previous protocols. This signal was fully specific for the senescence phenotype (i.e. absent in vehicle treated cells), devoid of background and required lower exposure times (~120 milliseconds) to be detected with a regular fluorescence microscope. Of relevance, no fluorescence signal was detected neither from vehicle nor BMS-470539 treated cells that were stained for SA-β-Gal activity (S3 Fig), highlighting that fluorescence detection represents a unique feature of SBB-stained lipofuscin aggregates.

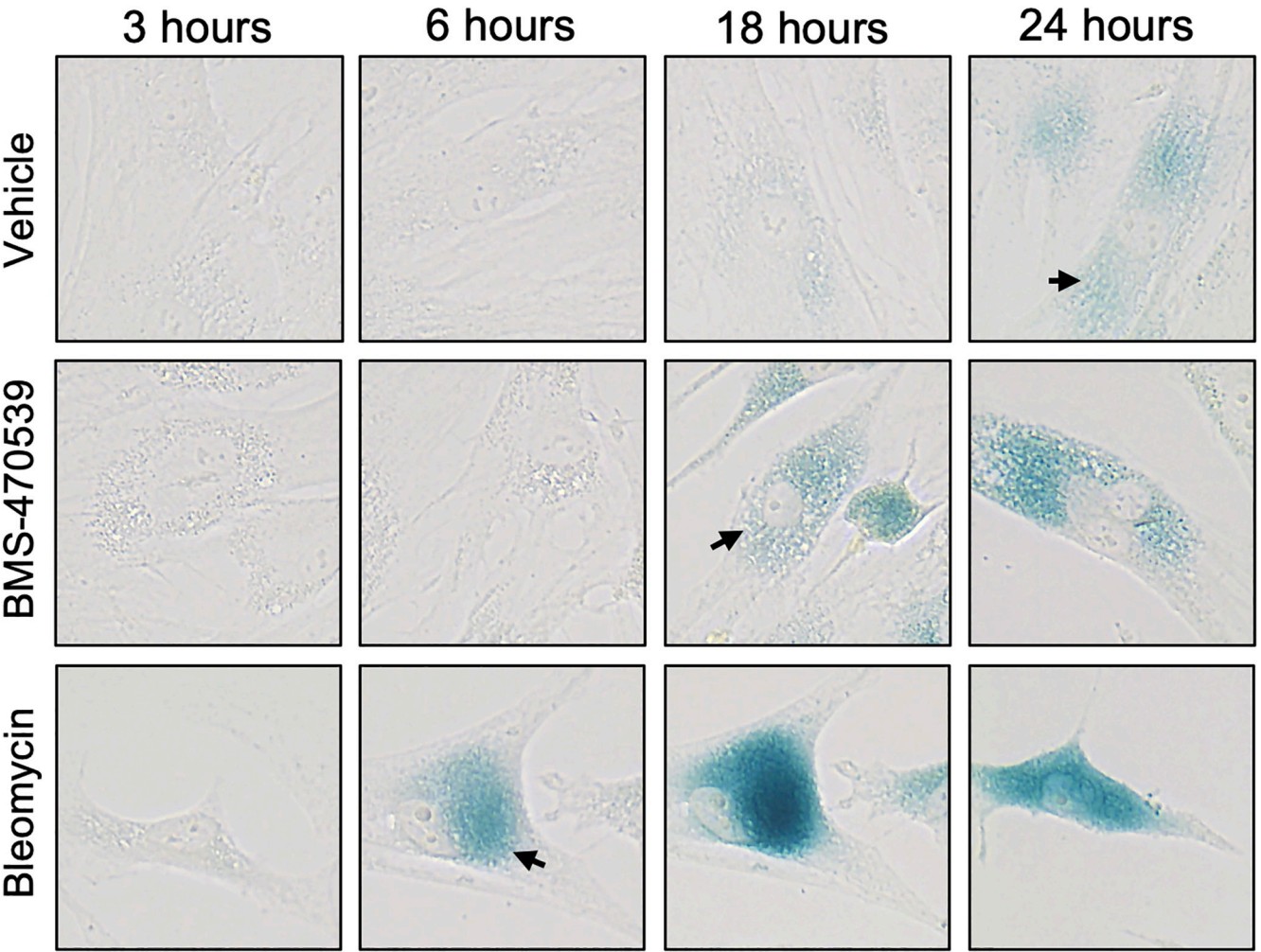

**Fig 3. Comparison of incubation times required by different stimuli for the detection of SA-β-Gal.** Cells were treated with vehicle (PBS), 10μM BMS-470539 or 3.3μM bleomycin every other day for 6 days and then stained for SA-β-Gal activity at different time points during the incubation period with the substrate X-Gal (3–24 hours). Images were captured using an EVOS XL Core Imaging System at 40X magnification. Arrows indicate positive detection of SA-β-Gal activity.

## Quantification of SBB stained cells is strongly associated with SA-β-Gal staining and the degree of senescence

To study the association between both senescence markers (SA-β-Gal and lipofuscin), two sets of cells were seeded in separate plates and stained using the SA-β-Gal staining kit (for brightfield microscopy analysis) or the SBB staining protocol presented here (for brightfield and fluorescence microscopy analyses). In all three cases, cells were treated with BMS-470539 at 1, 5, 10 or 20μM, or with vehicle (PBS) to determine the concentration-response effect of both markers. A clear concentration-dependent pro-senescence effect of the drug was observed using the three methods, all quantified as percentage of positive cells: SA-β-Gal staining using brightfield microscopy, SBB staining using brightfield microscopy and SBB staining using fluorescence microscopy (Fig 5A–5C). Senescence was detectable by the three methods from 1μM and reached close to 100% response at 10μM. Representative images are provided in Fig 5D.

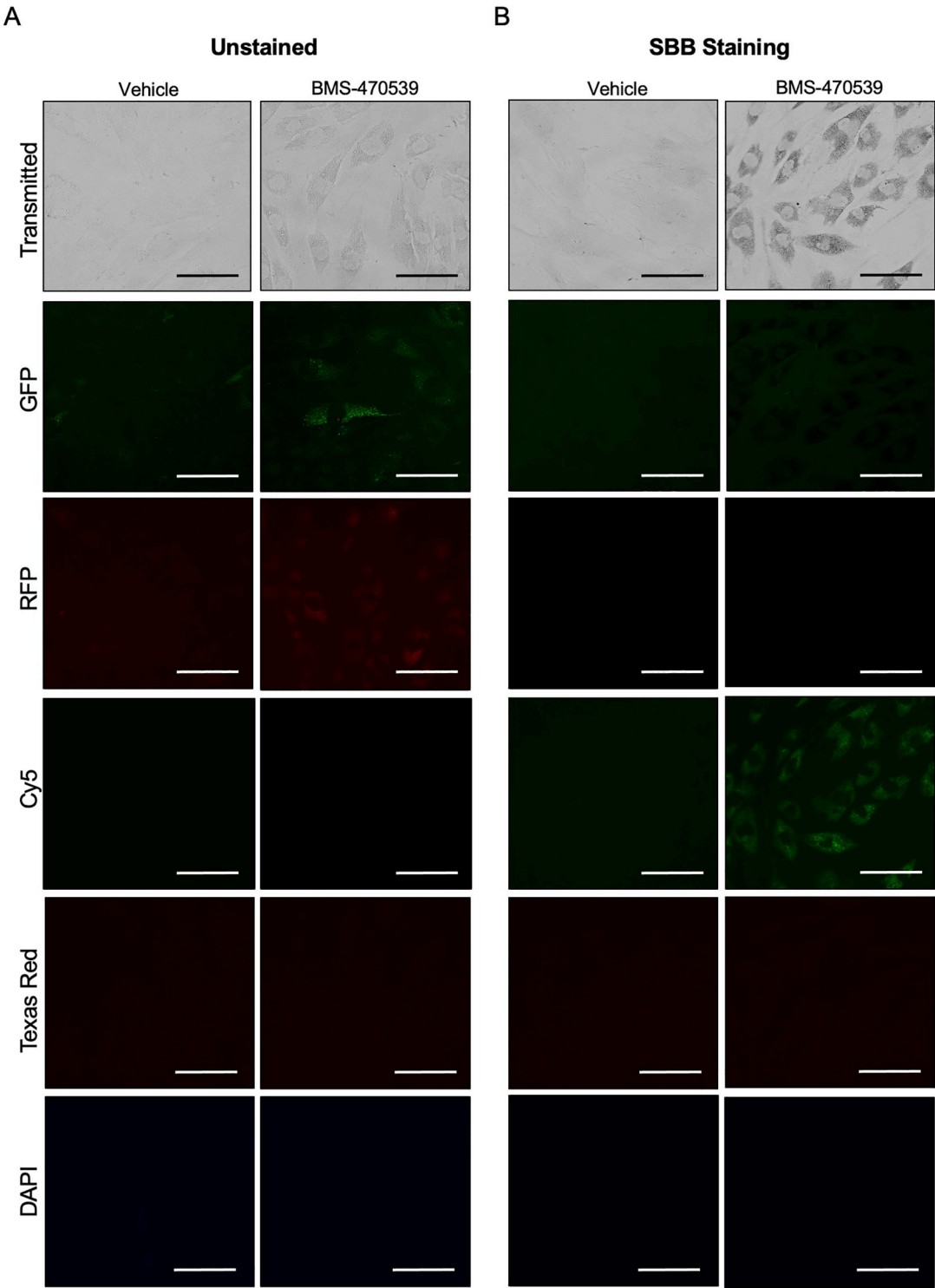

**Fig 4. Detection of lipofuscin using fluorescence microscopy.** Human dermal fibroblasts were treated with vehicle (PBS) or 20μM BMS-470539 for 6 days and the fluorescence signal determined at multiple channels before staining with SBB (A, denoting natural autofluorescence) and after staining with SBB (B, indicating positive detection of lipofuscin). Images were captured at 20X magnification using the EVOS FL Imaging System using the following Light Cubes: GFP (green, EX470/22—EM525/50), RFP (orange EX531/40—EM593/40), Texas Red (red, EX585/29—EM628/32), Cy5 (deep red, EX628/40—EM685/40), DAPI (blue, EX357/44—EX447/60) and transmitted channel. Scale bars represent 20μm.

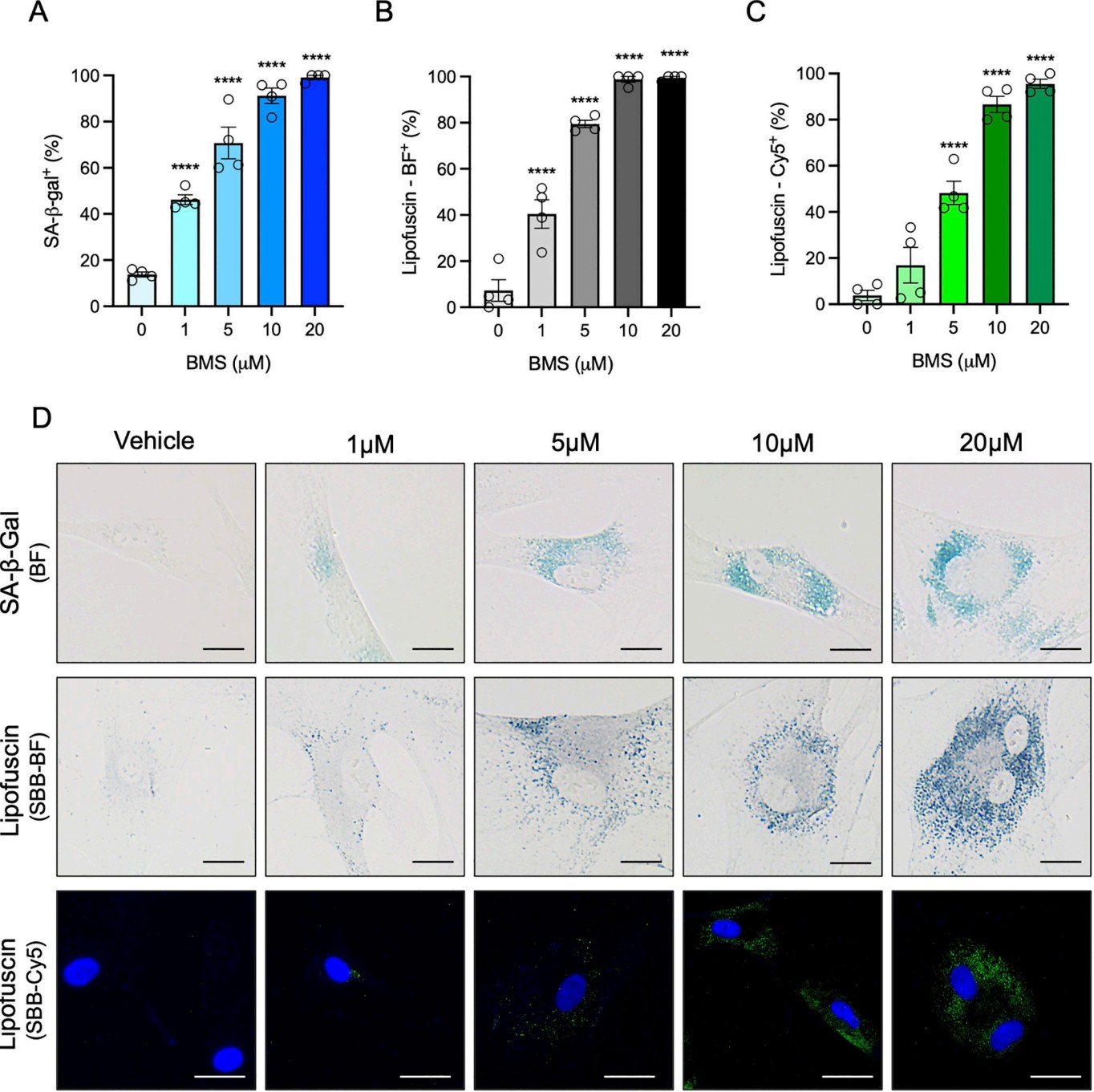

**Fig 5. Senescence quantification using the Sudan black B staining and SA-β-galactosidase assay.** Human dermal fibroblasts were treated with vehicle (PBS) or BMS-470539 (1–20μM) for 6 days. Cells were assessed by histochemical determination of SA-β-Gal (A), detected by brightfield (BF) microscopy, and of lipofuscin by staining with Sudan Black B dye, which was detected by brightfield microscopy (B) and by fluorescence microscopy (Cy5 channel, co-stained with DAPI for nuclear detection, C). The percentage of cells showing positive staining for SA-β-Gal activity and lipofuscin were quantified for each staining technique. A minimum of 20 cells were analysed per technical replicate. Data represent the mean ± SEM (n = 4; One-way ANOVA with multiple comparisons correction vs. vehicle; ****p<0.0001). Representative images are shown in panel D. Brightfield images were captured using an EVOS XL Core Imaging System at 40X magnification and fluorescence images captured using an EVOS™ FL Imaging System at 20X magnification. Scale bars represent 25μm.

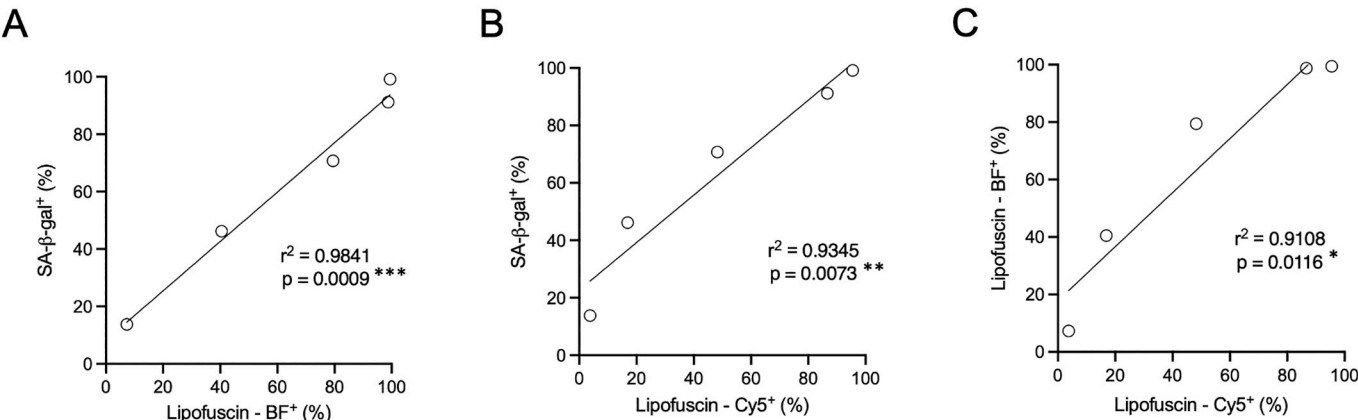

**Fig 6. Association between SA-β-galactosidase assay and Sudan black B staining.** Correlation analysis between each of the three staining techniques was conducted in order to ascertain the degree of concordance between the different protocols: A) SA-β-Gal versus lipofuscin detected by brightfield microscopy (BF), B) SA-β-Gal versus lipofuscin detected by fluorescence microscopy (Cy5), C) lipofuscin detected by brightfield microscopy (BF) versus lipofuscin detected by fluorescence microscopy (Cy5). The data used for this analysis correspond to the results presented in Fig 5 in which cells were treated with vehicle or with 1, 5, 10 or 20 μM BMS-470539). Data represent the mean of n = 4. Pearson's correlation test was used to obtain the coefficient of determination ($r^2$) and statistical significance, *p<0.05, **p<0.01, ***p<0.001.

In order to determine the level of agreement, the association across the three methods was addressed by analysing the correlation (Pearson's test) between them. As shown in Fig 6A and 6B, SA-β-Gal levels highly correlate with lipofuscin levels using SBB staining detected by both brightfield ($r^2 = 0.9841$) and by fluorescence ($r^2 = 0.9345$) microscopy. In addition, detection of lipofuscin by brightfield microscopy (Fig 6C) strongly correlates with its detection by fluorescence methods ($r^2 = 0.9108$). These results validate the use of lipofuscin detection methods to determine the induction of cellular senescence in cells cultured *in vitro*. Furthermore, the possibility of using fluorescence-based detection of lipofuscin aggregates offered by this SBB staining protocol has the advantage of allowing quantification not only as percentage of positive cells, but also by determining the intensity of the signal in each cell, as exemplified in S4 Fig.

## SBB can be co-stained with antibodies for fluorescence detection in cultured cells

Having demonstrated that SBB-stained lipofuscin is easily detectable using fluorescence microscopy, we next investigated if this technique could be used in conjunction with antibody-based immunofluorescence techniques, so that one could investigate proteins of interest and lipofuscin accumulation simultaneously. Critically, antibody-based fluorescence techniques cannot be used concurrently with SA-β-Gal staining, presenting a strong limitation in the co-detection of multiple markers of senescence or other biological processes. To test that possibility, cells were treated with 10μM BMS-470539 every other day for 6 days, after which the expression of alpha smooth muscle actin (αSMA) was determined using a standard immunofluorescence protocol (Fig 7A). After visualization, αSMA-stained cells were further stained with SBB solution using the method described in this work (starting from step 8 -incubation in Sudan Black B solution) and re-imaged using a fluorescence microscope (Fig 7B). This data confirms that antibody-stained α-SMA is preserved after applying the SBB staining protocol, demonstrating that both techniques can be used in conjunction. This is a significant improvement in the detection of markers of cellular senescence compared to standard SA-β-Gal

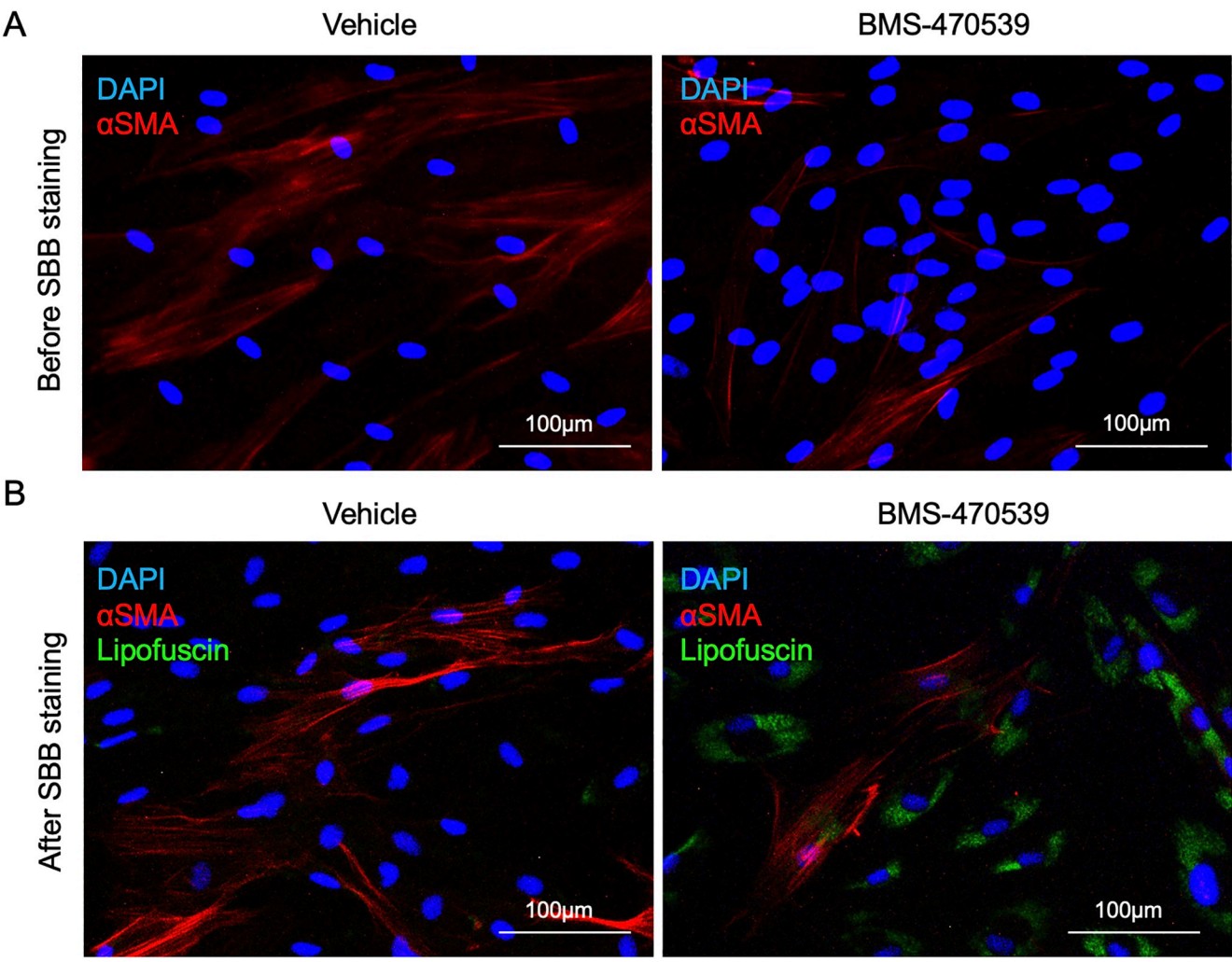

**Fig 7. Co-staining of Sudan black B in conjunction with immunofluorescence techniques.** BMS-470539 (10μM) and vehicle treated cells were probed for alpha-smooth muscle actin (α-SMA) expression using a standard immunofluorescence protocol, and counterstained with the nuclear marker DAPI. Images were then captured using the EVOS FL Imaging System with the Texas Red channel at 20X magnification (A). Cells when then stained using the SBB staining protocol and visualised using the EVOS FL Imaging System using the Texas Red (αSMA) and Cy5 (lipofuscin) channels, at 20X magnification (B). Scale bars represent 100μm.

staining. A comparison of the different aspects and advantages of the methodology described here based on lipofuscin staining with SBB, with the use of SA-β-Gal staining for the detection of cellular senescence in presented in Fig 8.

## Discussion

Lipofuscin has been described as a marker of cellular senescence for over two decades [17] with the first detailed report of its detection using Sudan black B staining in cultured cells published in 2013 [16]. However, lipofuscin staining, in contrast to SA-β-Gal detection, has rarely been used as a method to detect cellular senescence in cultured cells. Furthermore, when used, it has been demoted to a qualitative marker only, and a method to quantify lipofuscin accumulation has not been described to date. Here, we confirm that lipofuscin, detectable *via* SBB staining, is a pan-marker of senescence, and its accumulation is particularly pronounced after

| | SA-β-Galactosidase | Lipofuscin |
|---|---|---|
| | **Enzymatic hydrolysis of X-Gal** | **Sudan Black B staining** |
| **Senescence pan-marker** | Yes | Yes |
| ***In vitro* cell cultures** | Requires freshly fixed cells | Fixed for hours/days prior staining |
| **FFPE tissues** | Not suitable | Yes |
| **Optimization** | Staining time, cell type and specific conditions | Not required |
| **Quantification** | As % positive cells | As % and expression level / cell |
| **Microscopy detection** | Brightfield only | Brightfield, fluorescence and autofluorescence |
| **Antibody co-staining** | No | Yes |
| **Total protocol time** | Up to 24 hours | 1 hour |
| **Special equipment** | Incubator 37°C with no $CO_2$ (or sealed plates) | Plate shaker |
| **Cost** | ££ | £ |

**Fig 8. Comparison between the Sudan black B and the SA-β-galactosidase assays for the detection of cellular senescence.** Several aspects representing advantages and disadvantages for the SA-β-galactosidase activity determination and the lipofuscin staining using the protocol that we describe here are shown.

treatment of fibroblasts with the melanocortin compound BMS-470539. Whilst lipofuscin is known to exhibit autofluorescence in the green channel (FITC/GFP) [11], this autofluorescence is quenched when lipofuscin is stained with SBB [22, 23]. However, whilst the GFP signal is lost, we demonstrate here that a pronounced emission of signal in the far-red channel that is indeed derived from the Sudan black B-stained lipofuscin emerges. Thus, the fluorescence detection of SBB stained lipofuscin not only increases the sensitivity of the technique compared to brightfield detection, but also allows for the easy quantification using common image processing tools, as we show in this report.

There is no universal marker of senescence, but the most commonly used method to identify senescent cells is based on the detection of SA-β-Gal activity [24]. However, this technique requires comprehensive and context-dependent optimisation prior to use, particularly regarding the incubation time of the cells in the X-Gal staining solution. Commercial SA-β-Gal staining kits recommend incubation times ranging from 1 to 24 hours; if the incubation time is too short, the presence of senescence may not be detected, but if the incubation period is too long, this can result in erroneous positive staining even in control vehicle-treated cells, and thus the incorrection identification of senescent cells. Indeed, this is a common issue with enzyme-based assays [25]. Comparison between the pro-senescence effects of particular compounds is therefore challenging too, either between the effects on different cell types or when comparing different pro-senescence compounds, as we showed here: while BMS-470539-induced senescence required ~18h for the staining to develop, bleomycin-induced senescence was easily detectable as early as 6h after the X-Gal solution was added. Furthermore, these timings may likely vary with the concentration used. The Sudan black B staining procedure, conversely, is conducted following a single protocol for all conditions, regardless of senescence-induing stimulus or cell type. Equally, SA-β-Gal staining must be conducted on freshly fixed cells, as it is based on the detection of enzymatic activity (not on the presence of the enzyme), and cannot

be used on paraffin embedded tissues, which is not the case with Sudan black B staining, as demonstrated previously by others [16]. In addition, Rizou *et al* reported the synthesis of a biotin-labelled SBB analogue (GL13) which allows the detection of lipofuscin in fluids [26].

In addition to the practical advantages of SBB- based detection of lipofuscin over measuring SA-β-Gal activity, we have demonstrated that Sudan black B staining can also be visualised using fluorescence microscopy. This means that i) the technique becomes quantitative, as lipofuscin can be determined both as the proportion of positive cells and the expression level per cell, and ii) detection of lipofuscin can be combined together with standard immunofluorescence techniques to measure other proteins of interest at the same time. Critically, these outcomes are not achievable with current SA-β-Gal staining protocols, highlighting unique opportunities of using SBB staining for the detection of senescent cells.

Overall, the data presented here provide evidence of the validity of the reported protocol, and also demonstrate that the use of Sudan black B staining for the detection of cellular senescence provides several advantages over other methods, including being a fast and easy procedure, which can be conducted at a lower cost and with a reduced need for optimization, and that allows the possibility of fluorescence quantification as well as use in combination with antibody-based immunofluorescence techniques. As it is widely recognised that the reliable detection of cellular senescence should not rely solely on one marker, the methodology presented here can help to address this issue, as lipofuscin can be easily detected at the cellular level, simultaneously with other known markers of senescence like p16$^{INK4A}$, p21$^{WAF-1}$, DNA damage indicators (e.g., γ-H2AX, TP53BP1), anti-apoptotic proteins, or many others.

## Supporting information

**S1 Fig. Effect of inclusion of shaking during the staining procedure.** Human dermal fibroblasts were treated with vehicle (PBS) or 10μM BMS-470539 for 6 days and then stained with Sudan black B. The method was tested with (A) and without (B) using an orbital shaker during the incubation of the cells with the SBB solution. Images were captured using an EVOS XL Core Imaging System at 40X magnification. Selected areas were zoomed-in to highlight cytoplasmic and extracellular regions to determine the level of background and presence of SBB dye precipitates.
(TIFF)

**S2 Fig. Lipofuscin staining using old Sudan black B solution.** Human dermal fibroblasts were treated with vehicle (PBS) or 10μM BMS-470539 for 6 days and then stained with Sudan Black B using the protocol described herein. In this case, the Sudan black B solution used was prepared 7 days prior staining instead of freshly prepared the night before as indicated in the protocol. Images were captured using an EVOS XL Core Imaging System at 40X magnification.
(TIFF)

**S3 Fig. Assessing the fluorescence of SA-β-Gal using fluorescence microscopy.** Human dermal fibroblasts were treated with vehicle (PBS) or 20μM BMS-470539 for 6 days and fluorescence signal determined at multiple channels before histochemical determination of SA-β-Gal activity. Images were captured using the EVOS FL Imaging System (ThermoFisher) with the following channels: GFP (green, EX470/22—EM525/50), RFP (orange EX531/40—EM593/40), Texas Red (red, EX585/29—EM628/32), Cy5 (far-red, EX628/40—EM685/40), DAPI (blue, EX357/44—EX447/60) and transmitted channels. Scale bars represent 200μm.
(TIFF)

**S4 Fig. Quantification of SBB-stained lipofuscin expressed as fluorescence intensity per cell.** A) To calculate the fluorescence signal per cell in the far-red channel, Cy5 images were imported into Fiji and the integrated density measure function used to measure the integrated density of the whole image (Analyze → Measure→ Integrated density). This value was then normalised by the number of cells in the image to calculate the fluorescence per cell. B) Human dermal fibroblasts were treated with vehicle (PBS) or increasing concentrations of BMS-470539 for 6 days and subjected to the SBB staining protocol. The fluorescence intensity per image was calculated as explained in A, and normalised by the number of cells in each image. Data represent the mean ± SEM (n = 4; One-way ANOVA with multiple comparison correction vs. vehicle, ***p<0.001).
(TIFF)

**S1 File. Step-by-step protocol.** This can also be accessed on protocols.io ([https://dx.doi.org/10.17504/protocols.io.x54v9yw91g3e/v1](https://dx.doi.org/10.17504/protocols.io.x54v9yw91g3e/v1)).
(PDF)

## Acknowledgments

We thank Professor David J Abraham (Centre for Rheumatology, University College London) for his guidance in fibroblast culture, and Professor Christopher Denton and Dr Shi-Wen Xu (Centre for Rheumatology, University College London) for kindly providing the human dermal fibroblasts used in this study.

## Author Contributions

**Conceptualization:** Trinidad Montero-Melendez.

**Formal analysis:** Camilla S. A. Davan-Wetton.

**Funding acquisition:** Trinidad Montero-Melendez.

**Investigation:** Camilla S. A. Davan-Wetton.

**Methodology:** Camilla S. A. Davan-Wetton.

**Project administration:** Trinidad Montero-Melendez.

**Supervision:** Trinidad Montero-Melendez.

**Visualization:** Camilla S. A. Davan-Wetton.

**Writing – original draft:** Trinidad Montero-Melendez.

**Writing – review & editing:** Camilla S. A. Davan-Wetton, Trinidad Montero-Melendez.

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
