## [Decision Letter · Decision Letter 0]

4 Jun 2024

PONE-D-24-18320An optimised protocol for the detection of lipofuscin, a versatile and quantifiable marker of cellular senescencePLOS ONE

Dear Dr. Montero-Melendez,

Thank you for submitting your manuscript to PLOS ONE. After careful consideration, we feel that it has merit but does not fully meet PLOS ONE’s publication criteria as it currently stands. Therefore, we invite you to submit a revised version of the manuscript that addresses the points raised during the review process.

I have sent you the manuscript for minor revisions since enough feedback has been gathered. These revisions focus on enhancing the references and the statistical analysis of the work.

We look forward to receiving your revised manuscript.

Kind regards,

Luca Pesce, Ph.D.

Academic Editor

PLOS ONE

Journal Requirements:

Reviewers' comments:

Reviewer's Responses to Questions

**Comments to the Author**

1. Does the manuscript report a protocol which is of utility to the research community and adds value to the published literature?

Reviewer #1: Yes

Reviewer #2: Yes

2. Has the protocol been described in sufficient detail?

To answer this question, please click the link to protocols.io in the Materials and Methods section of the manuscript (if a link has been provided) or consult the step-by-step protocol in the Supporting Information files.

The step-by-step protocol should contain sufficient detail for another researcher to be able to reproduce all experiments and analyses.

Reviewer #1: Yes

Reviewer #2: Yes

3. Does the protocol describe a validated method?

Reviewer #1: Yes

Reviewer #2: Yes

4. If the manuscript contains new data, have the authors made this data fully available?

Reviewer #1: Yes

Reviewer #2: Yes

**5. Is the article presented in an intelligible fashion and written in standard English?**

Reviewer #1: Yes

Reviewer #2: Yes

6. Review Comments to the Author

Reviewer #1: The manuscript presents an alternative and optimized protocol for senescence detection through lipofuscin pigments visualization in cultured cells using the well-known marker Sudan Black B. Furthermore, the authors demonstrated that this optimized method enables the simultaneous detection of lipofuscin (with Sudan Black B) and other proteins using a standard immunofluorescence technique, offering an advantage over traditional beta-galactosidase-based methods. Overall, the manuscript is well-written, the authors provide sufficient details to replicate the protocol and the conclusions align with the experimental results presented.

Some minor revisions are needed:

1. Although Sudan Black B is a well-established marker for lipofuscin, the authors proposed an optimized detection method that was tested only in human dermal fibroblasts. It would be beneficial to test the Sudan Black B staining procedure on other types of cells that can undergo senescence-induction treatment.

2. The correlation analysis significantly strengthens the protocol's reliability. However, it is unclear from the main text and figure legends what types of treatments were used for the analysis (BMS-470539 at 1, 5, 10, or 20 µM?).

3. As stated by the authors, the work aims to determine the technical reliability of the protocol; therefore, more details are needed about the data analysis in Figures 5 and 6. Besides the number of replicates, the authors should report the number of cells analyzed for each condition and experiment. Was the integrated density measured for each cell (after creating a binary mask) or for the whole image?

Reviewer #2: The manuscript describes an optimized methodology to stain lipofuscin as a marker of cellular senescence via Sudan black-B, which can be detected either through brightfield and fluorescence microscopy. The protocol that the authors describe represents an improvement with respect to the state-of-the-art since it is easier and allows for the simultaneous staining of additional markers through immunohistochemistry protocols. The results are clear and well-structured and therefore I recommend the work for publication.

The following are questions that arose while reading the manuscript and may suggest additional information or clarifications that could strengthen the paper.

1. My first comment refers to the entire manuscript, where I think some terms are not properly used. The authors refer to different fluorescence detection windows (in terms of wavelength) as GFP-, RFP-, Cy3-… channels, although none of these fluorescent proteins or dyes are used in the experiments. Commercial microscopes may name the set of filters and parameters of excitation and emission as GFP, RFP, Cy3 channels for simplicity, but I believe it is wrong to maintain this nomenclature in the paper. What the authors are referring to is indeed the detection window in which, for example, Cy3 can be detected and whose excitation and emission filters have been found to be able to detect a fluorescent signal also from Sudan black-B. I suggest to the authors to change the nomenclature or, otherwise, to state they will refer to the specific detection window as “GFP” or “RFP” channel for simplicity.

2. In the introduction, I suggest to add a few more references (e.g. line 47, 53, 59..).

3. In line 90-91, the word “stain” seems redundant.

4. Line 121: I would specify that the chemicals described are used to induce senescence. Moreover, since this manuscript proposes a methodology, I would give more details (e.g. is PBS titrated to a specific pH? Is 4% PFA diluted in PBS or distilled water?) together with providing the product codes.

5. Figure 4 is completely black, I guess the image should be similar to what is shown in Supp Fig S3. Something has maybe happened while uploading the figure?

6. Line 283-293: I don’t have clear if the correlation analysis has been performed on the same cells stained with SA-b-Gal and then with SBB, or if it has been done on two different plates.

7. Line 328: typo error: “this autofluorescence is quenched when lipofuscin IS stained with SBB”

8. Line 329: the term “replaced” is not correct. What you are describing is that SBB quenches the autofluorescence signal of the cells that can be normally visualized by setting the exc and em parameters as you do to detect GFP and RFP (that you refer to as GFP and RFP channels). In addition to this, you found out that lipofuscin bound to SBB itself emits fluorescence in the “Cy3 channel”, but these are two different effects of using SBB and the second one is not a replacement of the first one.

9. Line 391-392: in Figure2, the top panel is lipofuscin accumulation with SBB and the middle is SA-b-Gal, but in the legend top and middle are inverted.

10. Line 469: I am confused, in the images there are not A and B but in the legend there are.

7. PLOS authors have the option to publish the peer review history of their article (what does this mean?). If published, this will include your full peer review and any attached files.

Reviewer #1: No

Reviewer #2: No

---

## [Author Response · Author response to Decision Letter 0]

10 Jun 2024

Manuscript: PONE-D-24-18320

Rebuttal Letter

Journal Requirements:

We have now revised and amended our manuscript according to the guidelines above.

The ORCID iD for the corresponding author has now been validated in Editorial Manager.

We have now deleted the ethics statement that we initially included in a section after the figure legends. 

We have revised the reference list and to the best of our knowledge, no retracted papers have been included. References are provided in Vancouver style, as per guidelines.

Reviewer #1: 

The manuscript presents an alternative and optimized protocol for senescence detection through lipofuscin pigments visualization in cultured cells using the well-known marker Sudan Black B. Furthermore, the authors demonstrated that this optimized method enables the simultaneous detection of lipofuscin (with Sudan Black B) and other proteins using a standard immunofluorescence technique, offering an advantage over traditional beta-galactosidase-based methods. Overall, the manuscript is well-written, the authors provide sufficient details to replicate the protocol and the conclusions align with the experimental results presented.

Some minor revisions are needed:

1. Although Sudan Black B is a well-established marker for lipofuscin, the authors proposed an optimized detection method that was tested only in human dermal fibroblasts. It would be beneficial to test the Sudan Black B staining procedure on other types of cells that can undergo senescence-induction treatment.

The suitability of SBB staining for the detection of lipofuscin as a marker of senescence in various cells and tissues has been extensively reported. The original paper (Georgakopoulou et al, 2013) describing lipofuscin accumulation as a general marker of senescence, already provided evidence of the suitability of SBB staining to detect lipofuscin in various cells, like mouse lung fibroblasts, two osteosarcoma cell lines, lung adenoma cells and prostate hyperplasia tissues. Evangelou et al (2017) report the use of SBB staining on human epithelial cells. Other examples include ovarian interstitial gland cells (Diaz-Hernandez,2022) or cardiomyocytes (Li et al, 2021). We have now included a new paragraph and these references in the introduction.

The scope of our work is to provide a step-by-step protocol optimised for the detection of lipofuscin in human fibroblasts. Our protocol can certainly be used as a starting point by other scientists who work on other cell types and other forms of senescence and provide optimization of timing and other settings which will likely be very cell-type and senescence-type specific, to be included in one single protocol as the one we report here.

2. The correlation analysis significantly strengthens the protocol's reliability. However, it is unclear from the main text and figure legends what types of treatments were used for the analysis (BMS-470539 at 1, 5, 10, or 20 µM?).

The correlation analysis provided in figure 6 correspond to the staining quantification reported in figure 5 (i.e. cells treated with 1, 5, 10 and 20 μM BMS-4700539, or vehicle). Only for the quantification of ‘bright field’ lipofuscin and ‘fluorescent’ lipofuscin, the same set of cells were used. However, as the colour visible under a bright field microscope for SA-βgalactosidase and SBB stain are very similar, separate sets of cells were used to allow precise visualization and quantification of each marker.

We have added a note to the legend for Figure 6 to clarify this point:

“[…] The data used for this analysis correspond to the results presented in Fig 5 in which cells were treated with vehicle or with 1, 5, 10, 20 μM BMS-470539). […].”

3. As stated by the authors, the work aims to determine the technical reliability of the protocol; therefore, more details are needed about the data analysis in Figures 5 and 6. Besides the number of replicates, the authors should report the number of cells analyzed for each condition and experiment. Was the integrated density measured for each cell (after creating a binary mask) or for the whole image?

In Figures 5 and 6, for each technical replicate, a minimum of 20 cells were analysed for every staining protocol. For measuring the intensity of the fluorescent SBB signal (S4 fig), the integrated density was calculated for the whole image and then normalised by dividing this value by the number of cells in the image. This information has been added to the figure legends (5, 6 and S4) and the methods section also clarified.

Reviewer #2: 

The manuscript describes an optimized methodology to stain lipofuscin as a marker of cellular senescence via Sudan black-B, which can be detected either through brightfield and fluorescence microscopy. The protocol that the authors describe represents an improvement with respect to the state-of-the-art since it is easier and allows for the simultaneous staining of additional markers through immunohistochemistry protocols. The results are clear and well-structured and therefore I recommend the work for publication.

The following are questions that arose while reading the manuscript and may suggest additional information or clarifications that could strengthen the paper.

1. My first comment refers to the entire manuscript, where I think some terms are not properly used. The authors refer to different fluorescence detection windows (in terms of wavelength) as GFP-, RFP-, Cy3-… channels, although none of these fluorescent proteins or dyes are used in the experiments. Commercial microscopes may name the set of filters and parameters of excitation and emission as GFP, RFP, Cy3 channels for simplicity, but I believe it is wrong to maintain this nomenclature in the paper. What the authors are referring to is indeed the detection window in which, for example, Cy3 can be detected and whose excitation and emission filters have been found to be able to detect a fluorescent signal also from Sudan black-B. I suggest to the authors to change the nomenclature or, otherwise, to state they will refer to the specific detection window as “GFP” or “RFP” channel for simplicity.

We understand the point raised and this is something we considered while writing the manuscript. As this is a methodological paper and the exact materials and equipment needs to be detailed, we decided that it was more correct to use the same terminology used by the manufacturer (ThermoFisher) to name their blocks (or Light Cubes, as they call them), so readers would easily find the detailed specifications if required. We did not choose to use “GFP” or “RFP” for simplicity, but because these are the names used by the manufacturer (perhaps the manufacturer used them for simplicity in the first place!). That is why we indeed provide the detection window of all the blocks that we use (e.g. GFP (green, EX470/22 - EM525/50)), so anyone would easily address the suitability of another manufacturer’s microscope blocks. Furthermore, for all the blocks, we also provide the “colour” name used for each one (e.g. green, far-red, etc), see methods:

“Cells were visualised using the EVOS FL Imaging System (ThermoFisher) at different channels depending on the experiments: Cy5 (far red, EX628/40 - EM685/40) was used to detect SBB-stained lipofuscin; Texas Red (red, EX585/29 - EM628/32) was used to detect Alexa Fluor 594 conjugated antibodies; DAPI (blue, EX357/44 - EX447/60) was used to detect

nuclei and the channels GFP (green, EX470/22 - EM525/50), RFP (orange EX531/40 -

EM593/40) were used to detect…”

2. In the introduction, I suggest to add a few more references (e.g. line 47, 53, 59..).

A number of additional references have now been included in the introduction. 

3. In line 90-91, the word “stain” seems redundant.

We have replaced “stain” with “detect”.

4. Line 121: I would specify that the chemicals described are used to induce senescence. Moreover, since this manuscript proposes a methodology, I would give more details (e.g. is PBS titrated to a specific pH? Is 4% PFA diluted in PBS or distilled water?) together with providing the product codes.

Further details have been provided and the catalogue numbers added to all reagents included in the manuscript. 

5. Figure 4 is completely black, I guess the image should be similar to what is shown in Supp Fig S3. Something has maybe happened while uploading the figure?

Yes, this happened because the figures provided to reviewers in the merged pdf built upon submission are in low resolution to make file size more manageable for reviewers. We noticed this issue and added a note to the editorial office during our submission.

6. Line 283-293: I don’t have clear if the correlation analysis has been performed on the same cells stained with SA-b-Gal and then with SBB, or if it has been done on two different plates.

This issue was also highlighted by Reviewer#1. Please see response to Question-2 in page 2.

7. Line 328: typo error: “this autofluorescence is quenched when lipofuscin IS stained with SBB”

This typo has been corrected.

8. Line 329: the term “replaced” is not correct. What you are describing is that SBB quenches the autofluorescence signal of the cells that can be normally visualized by setting the exc and em parameters as you do to detect GFP and RFP (that you refer to as GFP and RFP channels). In addition to this, you found out that lipofuscin bound to SBB itself emits fluorescence in the “Cy3 channel”, but these are two different effects of using SBB and the second one is not a replacement of the first one.

We have now removed the term replaced. This sentence now reads as follows:

“However, whilst the GFP signal is lost, we demonstrate here that a pronounced emission of signal in the far-red channel that is indeed derived from the Sudan black B-stained lipofuscin emerges.”

9. Line 391-392: in Figure2, the top panel is lipofuscin accumulation with SBB and the middle is SA-b-Gal, but in the legend top and middle are inverted.

This error has now been corrected.

10. Line 469: I am confused, in the images there are not A and B but in the legend there are.

This error has been corrected, the mention of A and B was mistaken.

---

## [Decision Letter · Decision Letter 1]

15 Jun 2024

An optimised protocol for the detection of lipofuscin, a versatile and quantifiable marker of cellular senescence

PONE-D-24-18320R1

Dear Dr. Montero-Melendez,

We’re pleased to inform you that your manuscript has been judged scientifically suitable for publication and will be formally accepted for publication once it meets all outstanding technical requirements.

Kind regards,

Luca Pesce, Ph.D.

Academic Editor

PLOS ONE

Additional Editor Comments (optional):

Reviewers' comments:

Reviewer's Responses to Questions

**Comments to the Author**

1. Does the manuscript report a protocol which is of utility to the research community and adds value to the published literature?

Reviewer #1: Yes

Reviewer #2: Yes

2. Has the protocol been described in sufficient detail?

To answer this question, please click the link to protocols.io in the Materials and Methods section of the manuscript (if a link has been provided) or consult the step-by-step protocol in the Supporting Information files.

The step-by-step protocol should contain sufficient detail for another researcher to be able to reproduce all experiments and analyses.

Reviewer #1: Yes

Reviewer #2: Yes

3. Does the protocol describe a validated method?

Reviewer #1: Yes

Reviewer #2: Yes

4. If the manuscript contains new data, have the authors made this data fully available?

Reviewer #1: Yes

Reviewer #2: Yes

**5. Is the article presented in an intelligible fashion and written in standard English?**

Reviewer #1: Yes

Reviewer #2: Yes

6. Review Comments to the Author

Reviewer #1: The authors have sufficiently answered my queries. I therefore have no further comments or revisions.

Reviewer #2: All my previous comments have been addressed properly, therefore I recommend publication of this revised version of the manuscript.

7. PLOS authors have the option to publish the peer review history of their article (what does this mean?). If published, this will include your full peer review and any attached files.

Reviewer #1: No

Reviewer #2: No

---

## [Editor Report · Acceptance letter]

4 Jul 2024

PONE-D-24-18320R1 

PLOS ONE

Dear Dr. Montero-Melendez, 

I'm pleased to inform you that your manuscript has been deemed suitable for publication in PLOS ONE. Congratulations! Your manuscript is now being handed over to our production team.

Kind regards, 

on behalf of

Dr. Luca Pesce 

Academic Editor

PLOS ONE